# CHEBMOE: A SPECTRAL-AWARE AND EXPERT-ADAPTIVE FRAMEWORK FOR GRAPH ANOMALY DETECTION

## ABSTRACT

Graph anomaly detection is critical for applications such as social networks, cybersecurity, and finance, yet remains challenging due to the unique spectral signatures of anomalies. In particular, anomalous nodes often exhibit high-frequency spectral patterns—a phenomenon known as *spectral shift*—which are easily suppressed by the low-pass nature of standard Graph Neural Networks (GNNs), resulting in *spectral washing* and poor anomaly detection. In this work, we present CHEBMOE, a novel and principled framework that directly addresses the spectral limitations of existing GNN-based anomaly detectors. Our key contributions are as follows: (1) We introduce a Chebyshev polynomial-based spectral feature extractor that efficiently preserves and amplifies high-frequency components, enabling the model to capture subtle spectral shifts associated with anomalies without requiring costly eigendecomposition. (2) We design a Mixture of Experts (MoE) anomaly detector with a learnable gating mechanism, allowing the model to adaptively aggregate diverse expert subnetworks and flexibly model complex anomaly patterns. (3) We propose a contrastive anomaly feature generator that leverages self-supervised contrastive learning to further enhance the discriminative power of node representations, improving robustness in the absence of labeled anomalies. Extensive experiments on seven real-world dynamic graph datasets demonstrate that CHEBMOE consistently outperforms state-of-the-art baselines. For example, it achieves ROC-AUC of 0.9906 on Wiki, 0.8791 on Reddit, and 0.9812 on UCI, along with consistently high F1-scores, effectively counteracting spectral washing and substantially advancing the state of graph anomaly detection.

## 1 INTRODUCTION

Graph data is ubiquitous in applications such as social networks, recommendation systems, bioinformatics, and financial monitoring. Detecting anomalies—nodes or subgraphs whose behavior significantly deviates from the majority—is critical for identifying faults, attacks, or irregular activities. This task is especially challenging in large-scale, dynamic graphs, where anomalies are rare, diverse, and evolving. Recent studies have revealed that anomalous nodes often exhibit stronger high-frequency spectral components than normal nodes, a phenomenon widelyknown as *spectral shift* (Tang et al., 2022). Intuitively, low-frequency signals capture smooth patterns shared by most nodes, while high-frequency signals encode abrupt variations indicative of anomalies. As formally analyzed in detail (see Appendix B.1), high-frequency components of graph signals correspond to rapidly varying Laplacian eigenmodes. Anomalous nodes are characterized by substantially higher high-frequency energy compared to normal nodes, making these spectral components crucial for distinguishing anomalies.

However, for modeling/encoding this graph data, widely used Graph Neural Networks (GNNs): including GCN, GAT, and GraphSAGE, inherently act as low-pass filters during message propagation. We observed that the GCN layer monotonically attenuates high-frequency components, leading to a *spectral washing* effect where the discriminative high-frequency signals of anomalous nodes are suppressed, making them harder to distinguish from normal nodes This phenomenon is illustrated in Figure 1 (see Appendix B.3 for more details), which shows that the normal nodes concentrate in low-frequency bands, anomalous nodes shift rightward, and GCN low-pass filtering (green dashed

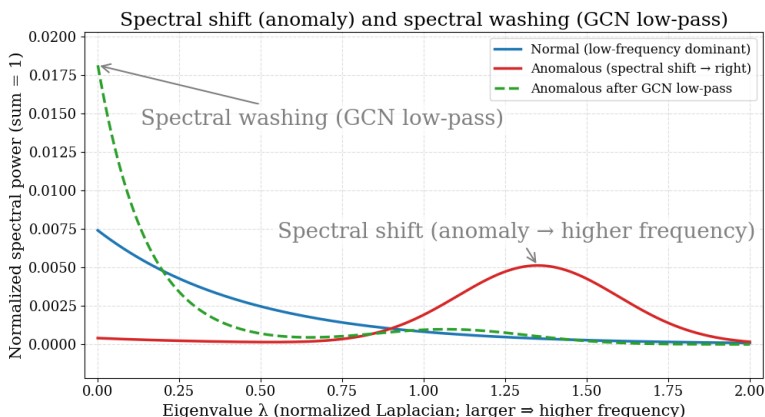

Figure 1: Spectral shift and washing in GNNs. X-axis: normalized Laplacian eigenvalue $\lambda$ (higher = higher frequency); Y-axis: normalized spectral power. Blue: normal (low-frequency); red: anomalous (high-frequency shift); green dashed: anomalous after GCN low-pass, with reduced but persistent high-frequency energy.

line) suppresses high-frequency components, making anomalies harder to detect. Existing graph anomaly detection methods, such as DOMINANT Ding et al. (2019), AnomalyDAE Fan et al. (2020), SAD Tian et al. (2023), and MAMF Hong et al. (2025), largely rely on GNNs for representation learning. While achieving promising results, they still inherit the low-pass bias, limiting their ability to capture the high-frequency anomaly signals revealed by spectral shift. A more rigorous and formal analysis of relevant concepts—including Dirichlet energy, graph Fourier decomposition, and the high-frequency energy of node features—is presented in Appendix B. There, we provide a theoretical examination of the limitations of GNNs, demonstrating how GNN propagation systematically attenuates high-frequency anomaly signals and thus hinders effective anomaly detection.

Addressing the spectral washing phenomenon in graph anomaly detection presents two primary challenges. First, standard GNNs inherently smooth graph signals, which leads to the suppression of high-frequency anomaly cues that are essential for identifying subtle or rare anomalies. Second, anomalies in real-world graphs are often temporally dynamic: anomalous nodes can exhibit varying high-frequency patterns over time, and novel anomaly types may arise that were not present during training. As a result, a single fixed-parameter model is insufficient to capture the full diversity of potential anomaly signals. This necessitates the development of adaptive approaches capable of dynamically responding to evolving anomaly patterns.

To simultaneously address the challenges of spectral washing and evolving anomaly patterns in graph data, we introduce a unified framework: **Cheb**yshev-based Network with **M**ixture-**o**f-**e**xperts (CHEBMOE) built on three synergistic components: **(1) Chebyshev Feature Extractor**, to preserve the high-frequency signals that are often indicative of anomalies, we replace standard GNN convolutions with a Chebyshev polynomial-based spectral filter. Chebyshev filtering leverages high-order polynomials of the graph Laplacian, allowing selective preservation of high-frequency signals while still aggregating local neighborhood information. This approach is theoretically justified in Appendix C, where we formally show how high-frequency anomaly signals are maintained. **(2) Contrastive Anomaly Feature Generator**, to address the scarcity of labeled anomalies, we use contrastive learning to enhance anomaly representations. By generating augmented views of anomalous node features through feature perturbation and training the encoder to maximize agreement between these views while distinguishing them from other nodes, we effectively synthesize additional anomaly samples and improve the robustness and generalization of learned features. **(3) Mixture-of-Experts (MoE) Anomaly Detector**, to handle diverse and evolving anomalies, we use a Mixture-of-Experts architecture. Multiple independently parameterized expert networks capture different high-frequency patterns, while a learnable gating network adaptively combines their outputs for each input. This enables flexible emphasis on various anomaly characteristics and improves generalization to unseen anomaly types. By integrating these components, CHEBMOE effectively preserves essential high-frequency anomaly signals, enriches the representation of scarce anoma-

lous data, and dynamically adapts to the emergence of new and evolving anomaly patterns. This holistic approach directly addresses the challenges posed by spectral washing and the limitations of static models, resulting in significantly enhanced anomaly detection performance on dynamic graphs. Comprehensive experiments across multiple benchmark datasets confirm that CHEBMOE achieves superior results compared to existing state-of-the-art methods.

We summarize our key contributions as follows:

- We rigorously identify and analyze the *spectral washing* phenomenon in graph anomaly detection, showing how standard GNNs suppress high-frequency components essential for detecting subtle anomalies (see Appendix B.1). That said, we provide both theoretical and practical insights into the limitations of existing GNN-based methods and motivates our spectral-preserving design.
- We propose CHEBMOE, a unified framework integrating (1) a Chebyshev polynomial-based feature extractor, (2) a contrastive anomaly feature generator, and (3) a Mixture-of-Experts (MoE) anomaly detector. This approach preserves high-frequency anomaly signals, augments limited anomaly data, and adaptively models diverse and evolving anomaly patterns—without prior knowledge of anomaly types.
- Extensive experiments on multiple dynamic graph anomaly detection benchmarks show that CHEBMOE consistently outperforms state-of-the-art baselines, validating the effectiveness of our spectral-preserving and adaptive framework.

## 2 RELATED WORK

We organize the related work into: (i) *Chebyshev polynomial filtering on graphs*; and (ii) *spectral shift and graph anomaly detection*.

**(i) Chebyshev Polynomial Filtering on Graphs.** Chebyshev polynomial filtering has been widely used for efficient graph signal processing and spectral analysis, enabling scalable computation on graphs without explicit eigen-decomposition Zhou (2010); Shuman et al. (2018); Chen et al. (2012); Sakiyama et al. (2016); Cheng et al. (2019); Liou et al. (2020); Huang et al. (2021); Tseng & Lee (2021); Du et al. (2024). Prior work applies Chebyshev filtering to eigenvalue problems Zhou (2010), distributed linear operators Shuman et al. (2018), graph wavelets Sakiyama et al. (2016), mesh augmentation Huang et al. (2021), and clustering Du et al. (2024), demonstrating its effectiveness in preserving high-frequency signals and capturing higher-order structure. We leverage Chebyshev filtering within a **ChebNet** framework to mitigate spectral washing and preserve anomaly-relevant components for dynamic graph anomaly detection.

**(ii) Spectral Shift and Graph Anomaly Detection.** Graph anomaly detection on dynamic graphs is challenging due to temporal evolution and heterogeneous anomaly patterns Xu et al. (2020); Ding et al. (2019); Xu et al. (2022); Tian et al. (2023); Hong et al. (2025). Most GCN-based models act as low-pass filters, oversmoothing features and suppressing high-frequency signals that are critical for detecting abrupt anomalies. Recent works partially address temporal dynamics or multi-view information Hong et al. (2025); Xu et al. (2022), but spectral washing remains a major issue.

We observe, the problem of spectral washing—where high-frequency anomaly signals are suppressed by standard GNN architectures—has been largely overlooked in prior work. Existing methods rarely address the preservation of these critical high-frequency components, leaving a significant gap in effective anomaly detection. In contrast, our framework, **CHEBMOE**, integrates Chebyshev filtering with a **Mixture of Experts (MoE)** detector and a contrastive anomaly generator to adaptively capture both low- and high-frequency patterns, improving robustness and generalization across diverse dynamic graphs.

## 3 CHEBMOE: **CHEB**YSHEV-BASED NETWORK WITH **M**IXTURE-**O**F-**E**XPERTS

In this section, we formally introduce the proposed framework: CHEBMOE for graph anomaly detection. We summarize the key notations used throughout this work in Appendix A (Table 3).

**Overview.** CHEBMOE targets dynamic graph anomaly detection by addressing two main challenges: (i) *spectral washing* in GNNs, which suppresses high-frequency anomaly signals, and (ii)

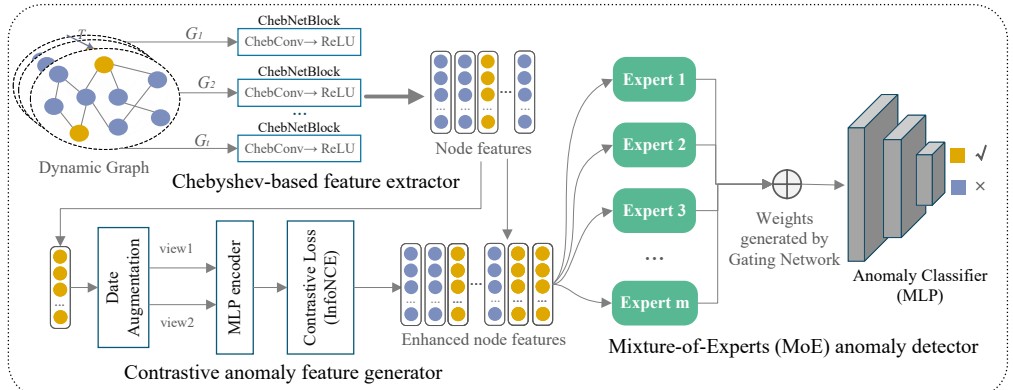

Figure 2: Overview of CHEBMOE. It comprises: (1) a Chebyshev Feature Extractor to retain high-frequency anomaly signals, (2) a Contrastive Anomaly Feature Generator to synthesize anomalous node representations, and (3) a Mixture-of-Experts Anomaly Detector to capture diverse anomaly patterns. We claim that this design mitigates spectral washing in GNNs and enhances detection of evolving anomalies.

shifting anomaly patterns over time. It encompasses: **(1) Chebyshev feature extractor**, for filtering in ChebNet to preserve high-frequency anomaly cues and reduce spectral washing. **(2) Contrastive anomaly generator**, for addressing anomaly scarcity, we introduce a contrastive generator that augments anomalies through view consistency, improving representation and robustness. **(3) Mixture-of-Experts detector**, for handling evolving anomalies, we employ a Mixture-of-Experts (MoE) detector with multiple experts (same architecture, independent parameters) and a gating network to combine their outputs, enabling diverse and adaptive detection.

### 3.1 CHEBYSHEV FEATURE EXTRACTOR

The Chebyshev feature extractor in CHEBMOE is designed to preserve both low-frequency (smooth) and high-frequency (anomaly-relevant) components in graph signals, thereby maintaining the spectral information crucial for effective anomaly detection. Unlike standard GNNs whose filters inherently decay over frequency and thus cause *spectral washing*, the Chebyshev polynomial filter allows flexible spectral shaping that explicitly retains high-frequency signals critical for detecting anomalies.

**Graph Laplacian and Spectral Filtering.** Given a graph $G = (V, E)$ with $N = |V|$ nodes, let $\mathbf{A} \in \mathbb{R}^{N \times N}$ denote the adjacency matrix, and $\mathbf{D}$ the diagonal degree matrix with $D_{ii} = \sum_j A_{ij}$. The normalized graph Laplacian is defined as:

$$\mathbf{L} = \mathbf{I}_N - \mathbf{D}^{-\frac{1}{2}} \mathbf{A} \mathbf{D}^{-\frac{1}{2}}, \tag{1}$$

where $\mathbf{I}_N$ is the identity matrix. Formally, let $\mathbf{L} = \mathbf{U} \mathbf{\Lambda} \mathbf{U}^\top$ be the eigendecomposition of $\mathbf{L}$ with $\mathbf{\Lambda} = \mathrm{diag}(\lambda_1, \ldots, \lambda_N)$ and orthonormal $\mathbf{U}$. Let the graph Fourier transform (GFT) of $\mathbf{X}$ be $\widehat{\mathbf{X}} = \mathbf{U}^\top \mathbf{X}$. A spectral filter with response $h : [0, \lambda_{\max}] \to \mathbb{R}$ acts as

$$\mathbf{Y} = h(\mathbf{L}) \mathbf{X} = \mathbf{U} h(\mathbf{\Lambda}) \mathbf{U}^\top \mathbf{X}, \quad h(\mathbf{\Lambda}) = \mathrm{diag}\big(h(\lambda_1), \ldots, h(\lambda_N)\big). \tag{2}$$

For first-order GCN propagation, one obtains an approximately low-pass response $h_{\mathrm{GCN}}(\lambda) = 1 - \lambda$ (see Appendix B.1 for details). Hence for any high-frequency band $\mathcal{H} = [\lambda_c, \lambda_{\max}]$ with $\lambda_c > 0$, the band energy contracts as:

$$\sum_{\lambda_i \in \mathcal{H}} \|\widehat{\mathbf{Y}}_{i,:}\|_2^2 = \sum_{\lambda_i \in \mathcal{H}} h_{\mathrm{GCN}}(\lambda_i)^2 \|\widehat{\mathbf{X}}_{i,:}\|_2^2 \leq (1 - \lambda_c)^2 \sum_{\lambda_i \in \mathcal{H}} \|\widehat{\mathbf{X}}_{i,:}\|_2^2, \tag{3}$$

This contraction of high-frequency energy is referred to as *spectral washing* (see Appendix B.1 for details). We argue that the Chebyshev polynomial formulation counteracts this by enabling controlled amplification or retention of high-frequency components, explained as follows:

**Chebyshev Polynomial Filter.** Let $\lambda_{\max}$ be the largest eigenvalue of $\mathbf{L}$ (for the normalized Laplacian, $\lambda_{\max} \approx 2$). The scaled Laplacian is:

$$\tilde{\mathbf{L}} = \frac{2}{\lambda_{\max}}\mathbf{L} - \mathbf{I}_N. \tag{4}$$

The Chebyshev polynomials $T_k(\cdot)$ are defined recursively as:

$$T_0(\tilde{\mathbf{L}}) = \mathbf{I}_N, \tag{5}$$
$$T_1(\tilde{\mathbf{L}}) = \tilde{\mathbf{L}}, \tag{6}$$
$$T_k(\tilde{\mathbf{L}}) = 2\tilde{\mathbf{L}}T_{k-1}(\tilde{\mathbf{L}}) - T_{k-2}(\tilde{\mathbf{L}}), \quad k \geq 2. \tag{7}$$

Given an input node feature matrix $\mathbf{X} \in \mathbb{R}^{N \times d}$, the Chebyshev spectral filtering of order $K$ is formulated as:

$$\mathbf{Y} = \sum_{k=0}^{K} T_k(\tilde{\mathbf{L}})\mathbf{X}\boldsymbol{\Theta}_k, \tag{8}$$

where $\boldsymbol{\Theta}_k \in \mathbb{R}^{d \times F}$ are learnable weight matrices for each polynomial order $k$, and $F$ is the output feature dimension. Equivalently in the spectral domain, the induced frequency response is

$$h_{\boldsymbol{\Theta}}(\lambda) = \sum_{k=0}^{K} \theta_k\, T_k\left(\frac{2\lambda}{\lambda_{\max}} - 1\right), \quad \text{with } \theta_k \text{ determined by } \boldsymbol{\Theta}_k. \tag{9}$$

By the Chebyshev minimax property, for any continuous target response $r^* : [0, \lambda_{\max}] \to \mathbb{R}$ and any $\varepsilon > 0$, there exists $K$ and coefficients $\{\theta_k\}_{k=0}^{K}$ such that

$$\sup_{\lambda \in [0,\lambda_{\max}]} \left|h_{\boldsymbol{\Theta}}(\lambda) - r^*(\lambda)\right| \leq \varepsilon, \quad \text{see Proposition 6 in Appendix C.} \tag{10}$$

The ChebNet layer produces the following output:

$$\mathbf{H} = \sigma(\mathbf{Y}) = \sigma\left(\sum_{k=0}^{K} T_k(\tilde{\mathbf{L}})\mathbf{X}\boldsymbol{\Theta}_k\right), \tag{11}$$

where $\sigma(\cdot)$ is a nonlinear activation function (e.g., ReLU).

**Implementation and Efficiency.** The Chebyshev filtering can be efficiently computed by recursively generating $T_k(\tilde{\mathbf{L}})\mathbf{X}$ for $k = 0, \ldots, K$, thus avoiding explicit eigen-decomposition and enabling scalability to large graphs; see Lemma 5 in Appendix C.

**Spectral Coverage and Anomaly Sensitivity.** By tuning the polynomial order $K$, ChebNet flexibly shapes its spectral response to balance low- and high-frequency components, as formalized in Proposition 6 and Corollary 7 (Appendix C). Specifically, smaller $K$ emphasizes smooth low-frequency patterns, while larger $K$ selectively retains sharp high-frequency variations that are often indicative of anomalies. This property directly mitigates the spectral washing effect commonly observed in standard GNNs. Beyond this intuitive perspective, we next formalize the high-frequency energy preservation guarantee. Moreover, let $\mathcal{H} = [\lambda_c, \lambda_{\max}]$ denote a high-frequency band and suppose the target $r^*$ satisfies $r^*(\lambda) \geq \gamma > 0$ for all $\lambda \in \mathcal{H}$. Combining (9)–(10) yields the energy preservation guarantee

$$\sum_{\lambda_i \in \mathcal{H}} \|\widehat{\mathbf{Y}}_{i,:}\|_2^2 \geq (\gamma - \varepsilon)^2 \sum_{\lambda_i \in \mathcal{H}} \|\widehat{\mathbf{X}}_{i,:}\|_2^2, \quad \text{formalized in Corollary 7 (Appendix C).} \tag{12}$$

Thus, with appropriately chosen $K$ and coefficients, ChebNet provably preserves anomaly-relevant high-frequency energy while allowing control over low-frequency gain. For dynamic graphs, we apply the ChebNet layer to each temporal snapshot $G_t$ with features $\mathbf{X}_t$, yielding spectral-aware node embeddings $\mathbf{H}_t$ for downstream anomaly detection. The Chebyshev feature extractor flexibly shapes the frequency response to preserve high-frequency anomaly signals while aggregating smooth graph information, directly addressing spectral washing. We claim, this principled approach underpins its effectiveness for both static and dynamic anomaly detection.

## 3.2 Contrastive Anomaly Feature Generator

To address the scarcity of anomaly samples, we introduce a *contrastive anomaly feature generator* that synthesizes diverse anomaly-like features by maximizing similarity between augmented views of the same anomaly and dissimilarity between different anomalies. This expands the anomaly feature space and improves representation robustness.

**Input and Feature Extraction.** Let $\mathbf{H} = \{\mathbf{h}_i \in \mathbb{R}^F\}_{i=1}^N$ be the node features from the Chebyshev extractor, where $N$ is the number of nodes and $F$ the feature dimension. We select features of nodes flagged as anomalous, which serve as input to the contrastive generator.

**Latent Embedding via MLP Encoder.** Each anomaly feature $\mathbf{h}_i$ is encoded into a latent vector $\mathbf{z}_i \in \mathbb{R}^p$ using a multi-layer perceptron (MLP):

$$\mathbf{z}_i = f_{\text{MLP}}(\mathbf{h}_i) = \mathbf{W}_3 \cdot \sigma\left(\mathbf{W}_2 \cdot \sigma\left(\mathbf{W}_1\mathbf{h}_i + \mathbf{b}_1\right) + \mathbf{b}_2\right) + \mathbf{b}_3, \tag{13}$$

where $\mathbf{W}_1, \mathbf{W}_2, \mathbf{W}_3$ and $\mathbf{b}_1, \mathbf{b}_2, \mathbf{b}_3$ are learnable parameters, and $\sigma(\cdot)$ is a nonlinear activation (e.g., ReLU). We use unit-norm embeddings $\tilde{\mathbf{z}}_i = \mathbf{z}_i / \|\mathbf{z}_i\|$, so cosine similarity reduces to a dot product.

**Contrastive Learning Objective.** The encoder is trained with the InfoNCE loss (Zhang et al., 2023). For a batch of $B$ anomaly samples, we generate two augmentations per node, yielding $2B$ views. Let $\{\tilde{\mathbf{z}}_k\}_{k=1}^{2B}$ be the normalized embeddings. For each positive pair $(i, j)$ (two views of the same node), the loss is:

$$\ell_{i,j} = -\log \frac{\exp\left(\tilde{\mathbf{z}}_i^\top \tilde{\mathbf{z}}_j / \tau\right)}{\sum_{\substack{k=1 \\ k \neq i}}^{2B} \exp\left(\tilde{\mathbf{z}}_i^\top \tilde{\mathbf{z}}_k / \tau\right)}. \tag{14}$$

Here $\tau > 0$ is a temperature parameter. The total loss is:

$$\mathcal{L}_{\text{contrastive}} = \frac{1}{2B} \sum_{k=1}^B \left[\ell_{2k-1,2k} + \ell_{2k,2k-1}\right]. \tag{15}$$

This contrastive approach learns embeddings where augmented views of the same anomaly are close, and different anomalies are distinct—without requiring explicit anomaly labels. The resulting diverse anomaly representations enhance downstream anomaly detection.

## 3.3 Mixture of Experts-based Anomaly Detector

To flexibly capture diverse and evolving anomaly patterns—especially high-frequency variations in dynamic graphs—we employ a Mixture of Experts (MoE) as our anomaly detector. A single model with fixed parameters struggles to handle the temporal and spectral heterogeneity of anomalies. Instead, our MoE uses multiple homogeneous experts with independent parameters, allowing each to specialize automatically during training, while a gating network adaptively combines their outputs.

**Expert Architecture.** Given input features $\mathbf{x} \in \mathbb{R}^{C \times L}$ ($C$: feature dimension, $L$: sequence length, typically $L = 1$), we instantiate $M$ experts $\{E_m\}_{m=1}^M$, each with identical architecture (e.g., MLP or CNN) but independent parameters. Each expert outputs:

$$\mathbf{e}_m = E_m(\mathbf{x}) \in \mathbb{R}^d \tag{16}$$

where $d$ is the expert output dimension. Experts naturally specialize in different anomaly patterns during training.

**Gating Network.** A gating network $G(\cdot)$ computes a probability distribution over experts for each input:

$$\boldsymbol{\alpha} = G(\mathbf{x}) = \text{softmax}\left(W_g \cdot \text{pool}(\text{Conv}(\mathbf{x})) + \mathbf{b}_g\right) \in \mathbb{R}^M \tag{17}$$

where $\text{Conv}(\cdot)$ is a convolutional layer, $\text{pool}(\cdot)$ is an adaptive pooling (e.g., global average pooling), and $W_g, \mathbf{b}_g$ are learnable parameters. The softmax ensures $\alpha_m \geq 0$ and $\sum_{m=1}^M \alpha_m = 1$.

**MoE Aggregation and Classification.** The final representation is a weighted sum of expert outputs, $\mathbf{z} = \sum_{m=1}^{M} \alpha_m \mathbf{e}_m$. This is passed to a classifier $C(\cdot)$ (e.g., MLP) to predict anomaly probability:

$$\hat{\mathbf{y}} = \text{softmax}(C(\mathbf{z})) \in \mathbb{R}^2 \tag{18}$$

where the output corresponds to normal and anomaly classes. The MoE detector is trained end-to-end by minimizing cross-entropy loss over a batch of $N$ samples:

$$\mathcal{L}_{\text{CE}} = -\frac{1}{N} \sum_{i=1}^{N} \sum_{c=1}^{2} y_{i,c} \log \hat{y}_{i,c} \tag{19}$$

where $y_{i,c}$ is the one-hot ground-truth label. In order to promote expert specialization and balanced usage, we add two regularizers: (1) a sparsity term based on gate entropy, and (2) a load-balancing term based on average gate usage. For a batch of size $N$, let $\boldsymbol{\alpha}^{(n)}$ be the gate for sample $n$, and $\bar{\boldsymbol{\alpha}} = \frac{1}{N} \sum_{n=1}^{N} \boldsymbol{\alpha}^{(n)}$. We define the following regularizers:

$$\mathcal{L}_{\text{spar}} = -\frac{1}{N} \sum_{n=1}^{N} \sum_{m=1}^{M} \alpha_m^{(n)} \log \alpha_m^{(n)} \tag{20}$$

$$\mathcal{L}_{\text{bal}} = \text{KL}\left(\bar{\boldsymbol{\alpha}} \,\|\, \mathbf{u}\right), \quad \mathbf{u} = (1/M, \ldots, 1/M) \tag{21}$$

The total loss is:

$$\mathcal{L}_{\text{total}} = \mathcal{L}_{\text{CE}} + \lambda_1 \mathcal{L}_{\text{spar}} + \lambda_2 \mathcal{L}_{\text{bal}} \tag{22}$$

with nonnegative coefficients $\lambda_1, \lambda_2$. Small $k$ and low gate entropy encourage specialization, while the balancing term prevents expert under-utilization. In summary, our MoE detector combines homogeneous experts and a gating network to adaptively specialize and robustly detect diverse, high-frequency anomaly patterns in dynamic graphs.

**Training workflow of CHEBMOE.** The training and inference of CHEBMOE follow four stages: (1) node features from each graph snapshot $G_t$ are filtered via Chebyshev polynomials to retain low- and high-frequency signals; (2) the contrastive anomaly generator augments anomaly features to enrich training and improve robustness; (3) embeddings are processed by multiple experts in the Mixture-of-Experts module, with a gating network producing a weighted aggregation; (4) the aggregated representation is fed to a classifier to predict anomalies, trained end-to-end with cross-entropy loss and optional contrastive objectives. This workflow clarifies module interactions; detailed algorithms and time complexity are in Appendix D and E.

## 4 EXPERIMENTS

### 4.1 EXPERIMENTAL SETTINGS

**Datasets.** We evaluate CHEBMOE on six dynamic graph anomaly detection benchmarks: Wikipedia (Kumar et al., 2019), Reddit (Kumar et al., 2019), EU-Core1/3 (Guo et al., 2022), AL-PHA (Kumar et al., 2016), and UCI (Zheng et al., 2019). Detailed description of the datasets are given in Appendix F.

**Baselines.** We compare with seven representative methods, including TGAT (Xu et al., 2020), DOMINANT (Ding et al., 2019), CONAD (Xu et al., 2022), SAD (Tian et al., 2023), and other widely used baselines. Further details about the baseline models are given in Appendix F.

**Evaluation Metrics.** For performance evaluation, we adopt four standard metrics: ROC-AUC, AUPR, Precision, and F1-score. Further details and mathematical formulations of the evaluation metrics are given in Appendix F.

**Setup.** Each dataset is split into five temporal segments: the first four for training and validation, and the last for testing. The embedding dimension is set to $k = 128$, with a batch size of 100. All experiments are conducted on an NVIDIA RTX 4060 GPU, and results are averaged over 20 runs. We use PyTorch 1.13.1 with CUDA 12.2. The Adam optimizer Zhang (2018) is adopted with an initial learning rate of $1 \times 10^{-3}$.

Table 1: Performance comparison on Wiki, Reddit, and UCI datasets.

| Method | Wiki | | | | Reddit | | | | UCI | | | |
|---|---|---|---|---|---|---|---|---|---|---|---|---|
| | ROC-AUC | Precision | F1-score | AUPR | ROC-AUC | Precision | F1-score | AUPR | ROC-AUC | Precision | F1-score | AUPR |
| TGAT | 0.7576 | 0.4991 | 0.4995 | 0.0251 | 0.6222 | 0.4995 | 0.4998 | 0.0020 | 0.9491 | 0.4971 | 0.4985 | 0.1258 |
| DOMINANT | 0.6707 | 0.0947 | 0.1022 | 0.0845 | 0.6101 | 0.2340 | 0.2281 | 0.1602 | 0.5233 | 0.1895 | 0.0805 | 0.1912 |
| DONE | 0.6486 | 0.0777 | 0.0839 | 0.0775 | 0.5690 | 0.2200 | 0.2144 | 0.1353 | 0.4993 | 0.2000 | 0.0850 | 0.1848 |
| CONAD | 0.6698 | 0.0947 | 0.1022 | 0.0845 | 0.6119 | 0.2360 | 0.2300 | 0.1602 | 0.5225 | 0.1895 | 0.0805 | 0.1913 |
| AnomalyDAE | 0.6706 | 0.0922 | 0.0996 | 0.0831 | 0.5666 | 0.2280 | 0.2222 | 0.1444 | 0.4912 | 0.1895 | 0.0805 | 0.1835 |
| SAD | 0.8641 | 0.4991 | 0.4995 | 0.0181 | 0.6880 | 0.4995 | 0.4998 | 0.0027 | 0.9223 | 0.4971 | 0.4985 | 0.1746 |
| MAMF | 0.9355 | 0.8757 | 0.8307 | 0.7507 | 0.7221 | 0.8082 | 0.6425 | 0.6053 | 0.9735 | 0.9760 | 0.9741 | 0.9725 |
| CHEBMOE(Ours) | **0.9906** | **0.8814** | **0.8503** | **0.7718** | **0.8791** | **0.8581** | **0.7923** | **0.7067** | **0.9812** | **0.9821** | **0.9806** | **0.9808** |

Table 2: Performance comparison on EU-Core1, EU-Core3, and ALPHA datasets.

| Method | EU-Core1 | | | | EU-Core3 | | | | ALPHA | | | |
|---|---|---|---|---|---|---|---|---|---|---|---|---|
| | ROC-AUC | Precision | F1-score | AUPR | ROC-AUC | Precision | F1-score | AUPR | ROC-AUC | Precision | F1-score | AUPR |
| TGAT | 0.4475 | 0.4972 | 0.4986 | 0.0057 | 0.5558 | 0.4944 | 0.4972 | 0.0980 | 0.7542 | 0.4682 | 0.4836 | 0.1427 |
| DOMINANT | 0.5282 | 0.2745 | 0.1162 | 0.2214 | 0.5856 | 0.2157 | 0.0995 | 0.2123 | 0.6823 | 0.6579 | 0.3289 | 0.4152 |
| DONE | 0.5138 | 0.3137 | 0.1328 | 0.2247 | 0.5535 | 0.1569 | 0.0724 | 0.2019 | 0.6843 | 0.6474 | 0.3237 | 0.4006 |
| CONAD | 0.5255 | 0.2745 | 0.1162 | 0.2199 | 0.5868 | 0.2157 | 0.0995 | 0.2128 | 0.6816 | 0.6737 | 0.3368 | 0.4107 |
| AnomalyDAE | 0.4993 | 0.2745 | 0.1162 | 0.2170 | 0.5835 | 0.1961 | 0.0905 | 0.2063 | 0.4962 | 0.0947 | 0.0474 | 0.1436 |
| SAD | 0.5361 | 0.4944 | 0.4972 | 0.0170 | 0.9080 | 0.4986 | 0.4972 | 0.1555 | 0.7574 | 0.4682 | 0.4836 | 0.1414 |
| MAMF | 0.9573 | 0.9516 | 0.9455 | 0.9258 | 0.9403 | 0.9017 | 0.8748 | 0.8381 | 0.9355 | 0.9400 | 0.9293 | 0.9320 |
| CHEBMOE(Ours) | **0.9665** | **0.9734** | **0.9714** | **0.9696** | **0.9611** | **0.9606** | **0.9557** | **0.9597** | **0.9517** | **0.9478** | **0.9447** | **0.9325** |

## 4.2 PERFORMANCE COMPARISON

We evaluate our method on seven real-world dynamic graph datasets against seven state-of-the-art baselines (Tables 1–2). Across all datasets, our method achieves top or highly competitive performance in ROC-AUC, Precision, F1-score, and AUPR, demonstrating strong generalization and robustness. On large-scale dynamic graphs like Wikipedia and Reddit, existing methods degrade due to over-smoothing, limited temporal modeling, or loss of high-frequency signals. In contrast, our model maintains high performance, e.g., ROC-AUC of 0.9906 on Wikipedia and 0.8791 on Reddit, surpassing the strongest baseline (MAMF). The improvements stem from three key innovations: (i) Chebyshev filtering preserves high-frequency anomaly signals; (ii) the SimCLR-based anomaly generator enhances feature robustness; (iii) the Mixture-of-Experts detector adapts to heterogeneous anomalies. Detailed dataset-specific results are in Appendix F.

## 4.3 ABLATION STUDY

We evaluate two components: the Chebyshev filter and the SimCLR generator. Replacing the Chebyshev kernel with GCN or GAT (`GCN+SimCLR+MoE`, `GAT+SimCLR+MoE`) leads to consistent performance drops (Fig. 3), with GCN worst, indicating that low-pass filters suppress anomaly-relevant high-frequency signals. Substituting SimCLR with a GAN-based generator (`ChebNet+GAN+MoE`) also hurts results, especially AUPR and F1 (Fig. 4), as GAN tends to memorise anomalies rather than enforce invariant features. Overall, both Chebyshev and SimCLR are indispensable for preserving discriminative spectra and ensuring robust generalisation.

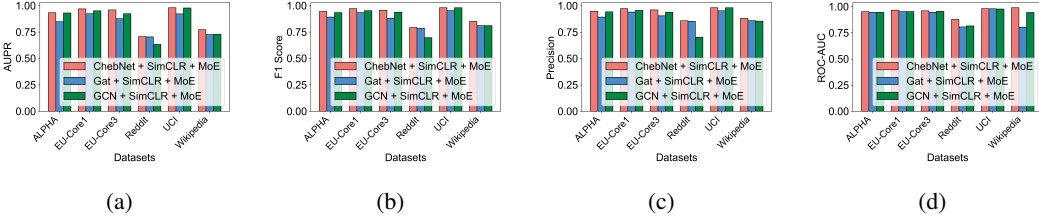

(a)  (b)  (c)  (d)

Figure 3: Group 1 ablation: performance across AUPR, F1, Precision, and AUC for the Chebyshev filter variants.

### 4.3.1 PARAMETER SENSITIVITY ANALYSIS

We study three key hyper-parameters: ChebNet order $K$, number of experts $M$, and SimCLR temperature $\tau$. Performance is generally stable across reasonable ranges: $K = 3$ balances accuracy and complexity; increasing $M$ slightly improves AUC; and $\tau$ values between 0.07–0.10 yield consistently strong results. Detailed results and full metric comparisons are deferred to Appendix H.

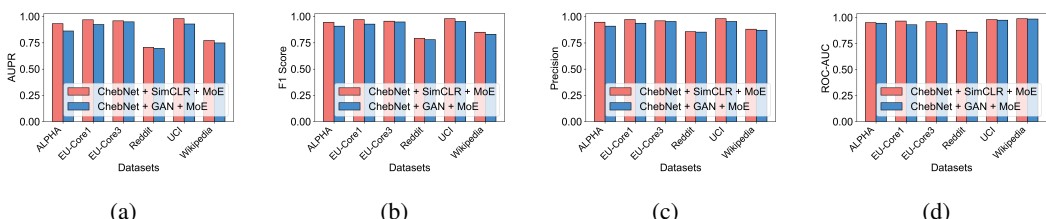

(a)            (b)            (c)            (d)

Figure 4: Group 2 ablation: performance across AUPR, F1, Precision, and AUC for the generator variants.

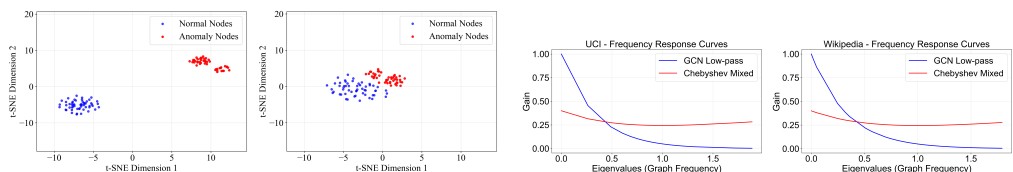

(a) t-SNE visualization of normal (blue) and anoma-   (b) Frequency response curves. Chebyshev filter-
lous (red) nodes. Left: with mitigation; Right: with-   ing preserves low- and high-frequency components;
out mitigation.   GCN filtering attenuates high-frequency signals.

Figure 5: Comparison of representation quality (t-SNE) and spectral behavior (frequency response).

## 4.4 T-SNE VISUALIZATION OF SPECTRAL WASHING MITIGATION

We visualize latent embeddings of 100 simulated anomalies and 100 normal nodes using t-SNE. Results on a representative dataset are shown in the main text, with others in Appendix G.2. As in Fig. 5a, when spectral washing is mitigated, anomalies form a clear cluster separated from normal nodes; without it, separation is weaker. This confirms that our generator produces high-quality anomalies and that preserving high-frequency components is critical for discrimination.

## 4.5 FREQUENCY RESPONSE ANALYSIS

We compare the frequency response of GCN propagation and our Chebyshev-based extraction. As shown in Fig. 5b, GCN suppresses high-frequency signals and obscures local anomalies, while our approach preserves both low- and high-frequency components, keeping anomalies distinguishable. Consistent patterns on other datasets (ALPHA, EU-CORE1/3, WIKIPEDIA, REDDIT) are provided in Appendix G.1, validating the effectiveness of our spectral preservation.

On the UCI and Wikipedia dataset (Figure 5b ), the Chebyshev filter maintains moderate gain across all frequency bands, ensuring that anomalies with high-frequency characteristics are preserved. In contrast, the GCN filter strongly suppresses high-frequency components, which can obscure anomalies. The analysis confirms that our Chebyshev-based approach effectively mitigates spectral washing. Similar trends are observed on other datasets (ALPHA, EU-CORE1, EU-CORE3, WIKIPEDIA, REDDIT), which are included in Appendix G.1.

## 5 CONCLUSION AND FUTURE WORK

We address node-level anomaly detection on dynamic graphs, focusing on the *spectral shift* problem, where standard GNNs suppress high-frequency anomaly signals. To solve this, we propose CHEB-MOE, which combines a Chebyshev spectral feature extractor, an anomaly generator, and a Mixture of Experts classifier. Our approach preserves crucial frequency information, improves anomaly representation, and adaptively models diverse anomalies. Experiments on benchmark datasets show that CHEBMOE consistently outperforms prior methods, especially for subtle and abrupt anomalies. Future work includes exploring advanced temporal modeling, enhancing cross-domain generalization, improving interpretability, and increasing robustness to adversarial attacks and noise.

## ETHICS STATEMENT

Our work focuses on graph anomaly detection, which has important applications in domains such as cybersecurity, fraud prevention, and infrastructure protection. While these applications can bring substantial societal benefits by improving security and stability, there are also potential risks if the technology is misused, such as infringing on individual privacy or enabling unauthorized surveillance through the inference of sensitive relationships or behaviors. We emphasize that the goal of this research is to advance the scientific understanding of graph representation learning and anomaly detection. We strongly advocate for responsible use of this technology, in compliance with ethical guidelines, data protection principles, and privacy regulations, and caution against applications that may cause harm to individuals or society.

## REPRODUCIBILITY STATEMENT

We have made extensive efforts to ensure the reproducibility of our work. All datasets used in our experiments are publicly available, and the main text provides detailed descriptions of data preprocessing, model architectures, hyperparameters, and training procedures. In addition, the code for our experiments is publicly released at `https://anonymous.4open.science/status/experiment-HL8530`, enabling other researchers to replicate and verify our results on the same datasets.

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

## A   NOTATION SUMMARY

We summarize the key notations used throughout this paper in Table 3.

Table 3: Notations

| Symbol | Description |
|---|---|
| $G = (V, E)$ | Graph with node set $V$ and edge set $E$ |
| $N = \|V\|$ | Number of nodes in graph $G$ |
| $\mathbf{A} \in \mathbb{R}^{N \times N}$ | Adjacency matrix of graph $G$ |
| $\mathbf{X} \in \mathbb{R}^{N \times d}$ | Input node feature matrix |
| $\mathbf{\Theta}_k \in \mathbb{R}^{d \times F}$ | Learnable weight matrix for polynomial order $k$ |
| $\mathbf{H} \in \mathbb{R}^{N \times F}$ | Output node embeddings from Chebyshev extractor |
| $f_{\mathrm{MLP}}(\cdot)$ | Multi-layer perceptron encoder in SimCLR module |
| $E_m(\cdot)$ | $m$-th expert network mapping input to embedding |
| $\boldsymbol{\alpha} = [\alpha_1, \ldots, \alpha_M]$ | Normalized expert weights, $\sum_m \alpha_m = 1$ |

## B   THEORETICAL ANALYSIS OF SPECTRAL WASHING

Here, we provide a rigorous and detailed theoretical analysis of the *spectral washing* phenomenon in standard GNNs, explaining why high-frequency anomaly signals are attenuated, and motivating the use of spectral-preserving filters such as Chebyshev polynomials.

### B.1   GRAPH SPECTRAL ANALYSIS

Consider a graph $G = (V, E)$ with $N$ nodes, adjacency matrix $\mathbf{A}$, and degree matrix $\mathbf{D}$. The normalized graph Laplacian is given by

$$\mathbf{L} = \mathbf{I}_N - \mathbf{D}^{-1/2}\mathbf{A}\mathbf{D}^{-1/2}, \tag{23}$$

which admits the eigendecomposition $\mathbf{L} = \mathbf{U}\mathbf{\Lambda}\mathbf{U}^\top$, where $\mathbf{\Lambda} = \mathrm{diag}(\lambda_1, \ldots, \lambda_N)$ and $0 = \lambda_1 \leq \cdots \leq \lambda_N \leq 2$.

A graph signal $x \in \mathbb{R}^N$ can be represented in the graph Fourier basis as

$$\hat{x} = \mathbf{U}^\top x, \tag{24}$$

where $\hat{x}$ contains the spectral coefficients.

Formally, consider the normalized graph Laplacian $\mathbf{L} = \mathbf{I}_N - \mathbf{D}^{-1/2}\mathbf{A}\mathbf{D}^{-1/2}$ with eigenpairs $(\lambda_i, \mathbf{u}_i)$, satisfying

$$\mathbf{L}\mathbf{u}_i = \lambda_i \mathbf{u}_i, \quad \|\mathbf{u}_i\|_2 = 1, \tag{25}$$

where $\mathbf{u}_i \in \mathbb{R}^N$ is the $i$-th unit-norm eigenvector and $\lambda_i$ its corresponding eigenvalue. The Laplacian quadratic form measures the smoothness of $\mathbf{u}_i$ over the graph:

$$\mathbf{u}_i^\top \mathbf{L} \mathbf{u}_i = \lambda_i = \frac{1}{2} \sum_{(p,q) \in E} \left( \frac{u_{i,p}}{\sqrt{d_p}} - \frac{u_{i,q}}{\sqrt{d_q}} \right)^2, \tag{26}$$

where:

- $(p, q) \in E$ denotes an edge connecting nodes $p$ and $q$,
- $u_{i,p}$ is the $p$-th component of eigenvector $\mathbf{u}_i$,
- $d_p$ is the degree of node $p$ (number of neighbors).

This decomposition shows that small eigenvalues $\lambda_i$ correspond to eigenvectors $\mathbf{u}_i$ that vary slowly across neighboring nodes (low-frequency, smooth modes), whereas large $\lambda_i$ correspond to eigenvectors with rapid variation across neighbors (high-frequency, localized modes), which often correspond to anomalies or sharp transitions.

Graph convolution operations can be viewed as spectral filtering. Let $x \in \mathbb{R}^N$ be a graph signal and $\hat{x} = \mathbf{U}^\top x$ its graph Fourier transform. Then a graph convolution with filter $g_\theta(\mathbf{L})$ produces
$$x' = g_\theta(\mathbf{L})x. \tag{27}$$
Applying the graph Fourier transform $\hat{x}' = \mathbf{U}^\top x'$ gives
$$\hat{x}' = \mathbf{U}^\top g_\theta(\mathbf{L})\mathbf{U}\hat{x} = g_\theta(\mathbf{\Lambda})\hat{x}, \tag{28}$$
where $g_\theta(\mathbf{\Lambda}) = \mathrm{diag}\big(g_\theta(\lambda_1), \ldots, g_\theta(\lambda_N)\big)$. In standard GCNs, $g_\theta(\lambda) \approx 1 - \lambda$, which monotonically decreases with $\lambda$, meaning that high-frequency components (large $\lambda_i$) are attenuated more than low-frequency components. Hence, a GCN layer acts as a low-pass filter, suppressing anomaly-relevant high-frequency signals.

## B.2 Node Signal Modeling

We represent a node feature vector $x \in \mathbb{R}^N$ as the sum of two components: a smooth background signal and a localized anomaly:
$$x = x_{\text{smooth}} + x_{\text{anom}}. \tag{29}$$
Here, $x_{\text{smooth}}$ represents the smooth background signal shared by most nodes, while $x_{\text{anom}}$ denotes localized deviations corresponding to anomalies.

Applying the graph Fourier transform yields
$$\hat{x} = \mathbf{U}^\top x, \tag{30}$$
where $\mathbf{U}$ is the eigenvector matrix of the graph Laplacian. By linearity,
$$\hat{x} = \hat{x}_{\text{smooth}} + \hat{x}_{\text{anom}}, \tag{31}$$
with $\hat{x}_{\text{smooth}} = \mathbf{U}^\top x_{\text{smooth}}$ and $\hat{x}_{\text{anom}} = \mathbf{U}^\top x_{\text{anom}}$.

**Frequency Decomposition.** To distinguish between smooth and anomalous components, we introduce a frequency threshold $\lambda_0$ and define spectral projectors:
$$\hat{x}_{\text{low}} = P_{\lambda \le \lambda_0}\hat{x}, \qquad \hat{x}_{\text{high}} = P_{\lambda > \lambda_0}\hat{x}, \tag{32}$$
where $P_{\lambda \le \lambda_0}$ and $P_{\lambda > \lambda_0}$ are diagonal matrices that select eigenvalues below or above $\lambda_0$, respectively. Specifically, the diagonal entries are 1 for selected eigenvalues and 0 otherwise. In the node domain, this gives
$$x = x_{\text{low}} + x_{\text{high}}, \quad x_{\text{low}} = \mathbf{U}\hat{x}_{\text{low}}, \ x_{\text{high}} = \mathbf{U}\hat{x}_{\text{high}}. \tag{33}$$

Using the Laplacian quadratic form (Section B.1),
$$x^\top \mathbf{L} x = \sum_{i=1}^{N} \lambda_i |\hat{x}_i|^2, \tag{34}$$
where $\hat{x}_i$ is the $i$-th spectral coefficient. For the smooth component $x_{\text{smooth}}$, $x_{\text{smooth}}^\top \mathbf{L} x_{\text{smooth}}$ is small, which implies that most of its energy $|\hat{x}_i|^2$ is concentrated at low eigenvalues $\lambda_i \le \lambda_0$ (low-frequency). For the anomalous component $x_{\text{anom}}$, the quadratic form is large due to abrupt variations, meaning its energy is concentrated at high eigenvalues $\lambda_i > \lambda_0$ (high-frequency). Hence, the decomposition
$$x = x_{\text{low}} + x_{\text{high}} \tag{35}$$
provides a theoretical justification linking smooth/normal signals to low-frequency modes and anomalies to high-frequency modes, since $x_{\text{smooth}}^\top \mathbf{L} x_{\text{smooth}}$ is small (energy concentrated in low frequencies) and $x_{\text{anom}}^\top \mathbf{L} x_{\text{anom}}$ is large (energy concentrated in high frequencies), so that $x_{\text{low}} \approx x_{\text{smooth}}$ and $x_{\text{high}} \approx x_{\text{anom}}$ in practice.

**High-Frequency Energy.** The energy in the high-frequency components is defined as

$$E_{\text{high}}(x) = \sum_{\lambda_i > \lambda_0} |\hat{x}_i|^2. \tag{36}$$

This energy measure corresponds to the Laplacian quadratic form, which quantifies the variation of the signal across connected nodes (see Section B.1 for derivation). Localized anomalies typically exhibit much higher high-frequency (Dirichlet) energy than smooth signals:

$$E_{\text{high}}(x_{\text{anom}}) \gg E_{\text{high}}(x_{\text{smooth}}). \tag{37}$$

This formalizes the intuition that anomalies correspond to sharp, localized changes, which appear as dominant high-frequency components in the spectral domain.

In summary, smooth (normal) node signals mainly occupy low-frequency modes, whereas localized anomalies primarily manifest in high-frequency modes. This provides a theoretical justification for using high-frequency energy as an indicator of anomalous nodes.

### B.3 GCN LAYER AS A LOW-PASS FILTER

A single GCN layer updates node features according to

$$x' = \hat{A}xW, \quad \hat{A} = \tilde{D}^{-1/2}\tilde{A}\tilde{D}^{-1/2}, \quad \tilde{A} = A + I, \tag{38}$$

where $A$ is the adjacency matrix, $I$ is the identity, and $W$ is a learnable weight matrix. Here, $\lambda_i$ is the $i$-th eigenvalue of the normalized Laplacian $\mathbf{L}$, which quantifies the smoothness of the corresponding Fourier mode $\mathbf{u}_i$.

In the spectral (Fourier) domain, this operation can be interpreted as applying a frequency-dependent filter:

$$\hat{x}_i' \approx h_{\text{gcn}}(\lambda_i)\hat{x}_i, \quad \text{where} \quad h_{\text{gcn}}(\lambda_i) = 1 - \lambda_i. \tag{39}$$

Here, $\lambda_i$ denotes the $i$-th eigenvalue of the normalized Laplacian. Since $h_{\text{gcn}}(\lambda_i)$ decreases as $\lambda_i$ increases, higher-frequency components (corresponding to larger $\lambda_i$) are increasingly suppressed. Thus, a GCN layer acts as a low-pass filter.

For a stack of $L$ GCN layers, the overall frequency response becomes

$$\hat{x}_i^{(L)} = (h_{\text{gcn}}(\lambda_i))^L \hat{x}_i = (1 - \lambda_i)^L \hat{x}_i. \tag{40}$$

This means that high-frequency components are attenuated exponentially with the number of layers $L$:

$$|\hat{x}_{\text{anom},i}^{(L)}|^2 = (1 - \lambda_i)^{2L} |\hat{x}_{\text{anom},i}|^2, \quad \forall \lambda_i > \lambda_0, \tag{41}$$

where $\lambda_0$ is a threshold separating low and high frequencies. For $\lambda_i > \lambda_0$, $(1 - \lambda_i)^{2L}$ quickly approaches zero as $L$ increases, so the high-frequency (anomaly-relevant) components are almost entirely washed out. Consequently, anomalies encoded in high-frequency components are strongly suppressed by GCN layers, which may reduce the model's sensitivity to localized anomalous nodes.

### B.4 ATTENUATION OF HIGH-FREQUENCY ENERGY

The total high-frequency energy after $L$ GCN layers is given by

$$E_{\text{high}}(x^{(L)}) = \sum_{\lambda_i > \lambda_0} |\hat{x}_i^{(L)}|^2 = \sum_{\lambda_i > \lambda_0} (1 - \lambda_i)^{2L} |\hat{x}_i|^2. \tag{42}$$

For normal (smooth) nodes, the high-frequency energy is already small and becomes negligible after propagation:

$$E_{\text{high}}(x_{\text{norm}}^{(L)}) \approx \sum_{\lambda_i > \lambda_0} (1 - \lambda_i)^{2L} |\hat{x}_{\text{smooth},i}|^2 \approx 0. \tag{43}$$

For anomalous nodes, which initially have large high-frequency energy, repeated GCN layers drastically reduce this energy:

$$E_{\text{high}}(x_{\text{anom}}^{(L)}) = \sum_{\lambda_i > \lambda_0} (1 - \lambda_i)^{2L} |\hat{x}_{\text{anom},i}|^2 \ll E_{\text{high}}(x_{\text{anom}}). \tag{44}$$

This shows that the distinctive high-frequency signature of anomalies is strongly suppressed.

The difference in high-frequency energy between anomalous and normal nodes after $L$ layers is

$$\Delta E_{\text{high}}^{(L)} = E_{\text{high}}(x_{\text{anom}}^{(L)}) - E_{\text{high}}(x_{\text{norm}}^{(L)}) \ll \Delta E_{\text{high}} = E_{\text{high}}(x_{\text{anom}}) - E_{\text{high}}(x_{\text{norm}}). \quad (45)$$

This quantifies the *spectral washing* effect: the gap in high-frequency energy that distinguishes anomalies from normal nodes is greatly diminished, making anomalies harder to detect.

## B.5 IMPLICATIONS FOR ANOMALY DETECTION

**Lemma 1** (Exponential suppression of high-frequency components by GCN). *Let $\lambda_0 \in (0, 2]$ denote the high-frequency threshold and let $L \in \mathbb{N}$. For a stack of $L$ GCN layers with frequency response $h_{\text{gcn}}(\lambda) = 1 - \lambda$, any spectral component $\hat{x}_i$ with $\lambda_i > \lambda_0$ satisfies*

$$\left|\hat{x}_i^{(L)}\right|^2 = \left(1 - \lambda_i\right)^{2L} \left|\hat{x}_i\right|^2 \leq \left(1 - \lambda_0\right)^{2L} \left|\hat{x}_i\right|^2.$$

*In particular, high-frequency components are suppressed at least geometrically in $L$.*

*Proof.* By Section B.3, the $L$-layer response is $(1 - \lambda_i)^L$ on eigenvalue $\lambda_i$. Since $\lambda \mapsto 1 - \lambda$ decreases on $[0, 2]$, for all $\lambda_i > \lambda_0$ we have $|(1 - \lambda_i)| \leq |(1 - \lambda_0)| < 1$. Squaring yields the claim. $\square$

**Proposition 2** (Geometric decay of high-frequency energy and gap). *Define $\gamma := \max_{\lambda_i > \lambda_0}(1 - \lambda_i)^2 < 1$. Then for any graph signal $x$ and any $L \in \mathbb{N}$,*

$$E_{\text{high}}(x^{(L)}) = \sum_{\lambda_i > \lambda_0} (1 - \lambda_i)^{2L} |\hat{x}_i|^2 \leq \gamma^L E_{\text{high}}(x).$$

*Moreover, if anomalies dominate pointwise in the high-frequency spectrum, i.e., $|\hat{x}_{\text{anom},i}|^2 \geq |\hat{x}_{\text{norm},i}|^2$ for all $\lambda_i > \lambda_0$, then the energy gap obeys*

$$\Delta E_{\text{high}}^{(L)} = E_{\text{high}}(x_{\text{anom}}^{(L)}) - E_{\text{high}}(x_{\text{norm}}^{(L)}) \leq \gamma^L \Delta E_{\text{high}}.$$

*Proof.* The first inequality follows by taking $\gamma$ outside the sum. Under pointwise dominance, each per-frequency difference is nonnegative and scaled by $(1 - \lambda_i)^{2L} \leq \gamma^L$, yielding the stated bound on the gap. $\square$

**Corollary 3** (Reduced separability under GCN propagation). *Under the conditions of the proposition, any anomaly detector whose score is a (strictly) increasing function of $E_{\text{high}}$ experiences a monotonically shrinking margin between anomalous and normal nodes. In particular, for any fixed threshold $\tau > 0$, there exists $L_\tau$ such that for all $L \geq L_\tau$, $\Delta E_{\text{high}}^{(L)} < \tau$.*

*Proof.* Immediate from $\Delta E_{\text{high}}^{(L)} \leq \gamma^L \Delta E_{\text{high}}$ with $\gamma \in (0, 1)$. $\square$

**Proposition 4** (Existence of spectral-preserving polynomial filters). *For any $\varepsilon > 0$ and constants $c \in (0, 1)$ and $\lambda_0 \in (0, 2]$, there exists a polynomial filter $h(\lambda)$ such that $|h(\lambda) - 1| \leq \varepsilon$ for all $\lambda > \lambda_0$ and $|h(\lambda)| \leq c$ for all $\lambda \in [0, \lambda_0]$. Such filters can be realized via Chebyshev polynomial expansions after scaling the spectrum to $[-1, 1]$.*

*Proof sketch.* Define a continuous target response $g(\lambda)$ that equals $1$ on $[\lambda_0 + \delta, 2]$ and is bounded by $c$ on $[0, \lambda_0 - \delta]$, with a smooth transition on $[\lambda_0 - \delta, \lambda_0 + \delta]$ for some small $\delta > 0$. By the Weierstrass approximation theorem (applied on the scaled interval), $g$ can be uniformly approximated by a polynomial $h$ within $\varepsilon$, yielding the stated properties. Implementing $h$ with Chebyshev polynomials gives an efficient, eigen-decomposition-free realization. $\square$

# C CHEBYSHEV FILTERS FOR PRESERVING HIGH-FREQUENCY INFORMATION

This section provides a clear theoretical explanation of how Chebyshev polynomial filters can counteract spectral washing in GCNs and preserve high-frequency spectral components that are crucial for effective anomaly detection.

## C.1 CHEBYSHEV POLYNOMIAL FILTERING

Let $\mathbf{L} = \mathbf{U}\mathbf{\Lambda}\mathbf{U}^\top$ denote the normalized graph Laplacian. A Chebyshev polynomial filter of order $K$ is defined as:

$$\mathbf{y} = \sum_{k=0}^{K} \theta_k T_k(\tilde{\mathbf{L}})\mathbf{x}, \tag{46}$$

where $\tilde{\mathbf{L}} = \frac{2}{\lambda_{\max}}\mathbf{L} - \mathbf{I}$ rescales the Laplacian so its eigenvalues lie in $[-1, 1]$, $T_k(\cdot)$ is the $k$-th Chebyshev polynomial, and $\theta_k$ are learnable filter coefficients.

In the spectral domain, this operation is equivalent to:

$$\hat{y}_i = \sum_{k=0}^{K} \theta_k T_k(\tilde{\lambda}_i)\hat{x}_i = h_{\text{cheb}}(\lambda_i)\hat{x}_i, \tag{47}$$

where $h_{\text{cheb}}(\lambda_i)$ is the frequency response of the filter at eigenvalue $\lambda_i$.

## C.2 PRESERVATION OF HIGH-FREQUENCY COMPONENTS

Chebyshev polynomials form an orthogonal basis on $[-1, 1]$, enabling the construction of filters that approximate arbitrary frequency responses. By appropriately choosing the coefficients $\theta_k$, we can design $h_{\text{cheb}}(\lambda)$ to satisfy:

$$h_{\text{cheb}}(\lambda_i) \approx 1, \quad \forall \lambda_i > \lambda_0, \tag{48}$$

where $\lambda_0$ is a chosen high-frequency threshold. This means that for high-frequency components (i.e., those with $\lambda_i > \lambda_0$), the filter passes them through with minimal attenuation:

$$\hat{x}'_{\text{anom},i} = h_{\text{cheb}}(\lambda_i)\hat{x}_{\text{anom},i} \approx \hat{x}_{\text{anom},i}, \quad \lambda_i > \lambda_0. \tag{49}$$

Thus, the spectral signatures of anomalies, which often reside in the high-frequency range, are preserved.

## C.3 RESTORATION OF HIGH-FREQUENCY ENERGY AND SEPARABILITY

After applying the Chebyshev filter, the high-frequency energy of the anomalous signal is:

$$E_{\text{high}}(x'_{\text{anom}}) = \sum_{\lambda_i > \lambda_0} |h_{\text{cheb}}(\lambda_i)\hat{x}_{\text{anom},i}|^2 \approx \sum_{\lambda_i > \lambda_0} |\hat{x}_{\text{anom},i}|^2 = E_{\text{high}}(x_{\text{anom}}). \tag{50}$$

This means the filter effectively restores the high-frequency energy that would otherwise be suppressed by standard GCN layers. Consequently, the difference in high-frequency energy between anomalous and normal nodes is also preserved:

$$\Delta E_{\text{high}}^{\text{cheb}} = E_{\text{high}}(x'_{\text{anom}}) - E_{\text{high}}(x'_{\text{norm}}) \approx \Delta E_{\text{high}}, \tag{51}$$

thereby mitigating the spectral washing effect and maintaining the separability needed for reliable anomaly detection.

## C.4 THEORETICAL IMPLICATIONS

**Lemma 5** (Eigen-free implementation via Chebyshev recurrence). *Let $\tilde{L} = \frac{2}{\lambda_{\max}}L - I$ and define $T_0 = I$, $T_1 = \tilde{L}$, and $T_k = 2\tilde{L}T_{k-1} - T_{k-2}$. Then for any feature matrix $X \in \mathbb{R}^{N \times F}$,*

$$\sum_{k=0}^{K} \theta_k T_k(\tilde{L})X$$

*can be computed without eigendecomposition using $K$ sparse matrix–vector multiplications per feature channel. Hence the implementation cost is linear in $K$ and in the number of nonzeros of $L$.*

**Proposition 6** (Uniform approximation of target frequency responses)**.** *Let $g : [0, 2] \to \mathbb{R}$ be continuous and let $h_K$ denote the degree-$K$ Chebyshev approximation of $g$ on $[0, 2]$ after spectrum scaling to $[-1, 1]$. Then $\|h_K - g\|_\infty \to 0$ as $K \to \infty$. In particular, for any $\varepsilon > 0$, there exists $K$ such that*

$$|h_K(\lambda) - 1| \le \varepsilon \ \text{for all } \lambda > \lambda_0, \qquad |h_K(\lambda)| \le c \ \text{for all } \lambda \in [0, \lambda_0]$$

*for some $c \in (0, 1)$, realizing a high-pass design that preserves high frequencies while controlling low-frequency gain.*

**Corollary 7** (Lower bound on preserved high-frequency energy)**.** *Suppose a filter $h$ satisfies $|h(\lambda) - 1| \le \varepsilon$ for all $\lambda > \lambda_0$. Then for any signal $x$,*

$$E_{\text{high}}(x') = \sum_{\lambda_i > \lambda_0} |h(\lambda_i)\hat{x}_i|^2 \ \ge \ (1 - \varepsilon)^2 \, E_{\text{high}}(x).$$

*If in addition $|h(\lambda)| \le c$ for all $\lambda \le \lambda_0$, then for anomalous and normal signals*

$$\Delta E'_{\text{high}} \ \ge \ (1 - \varepsilon)^2 E_{\text{high}}(x_{\text{anom}}) \, - c^2 E_{\text{high}}(x_{\text{norm}}),$$

*so the separability in high-frequency energy is maintained for small $\varepsilon$ and moderate $c$.*

The results above show that Chebyshev filters (i) admit efficient, eigen-free implementations, (ii) approximate desired high-pass responses, and (iii) provably preserve high-frequency energy and separability. This explains their effectiveness at counteracting spectral washing.

# D CHEBMOE—TRAINING ALGORITHM

---

**Algorithm 1** CHEBMOE Framework for Graph Anomaly Detection

---

**Require:** Dynamic graph snapshots $\mathcal{G} = \{G_t = (V_t, E_t, X_t)\}_{t=1}^T$

**Ensure:** Anomaly detection results $\{\hat{y}_v^{(t)}\}$ for nodes $v \in V_t$

1: **for** $t = 1$ to $T$ **do**
2:     Compute normalized graph Laplacian $\mathbf{L}^{(t)}$ (Eq. (1))
3:     Compute scaled Laplacian $\tilde{\mathbf{L}}^{(t)}$ (Eq. (4))
4:     Compute Chebyshev polynomial filters $T_k(\tilde{\mathbf{L}}^{(t)})$ for $k = 0, \ldots, K$ (Eq. (8))
5:     Extract spectral features $\mathbf{H}^{(t)} \leftarrow \sigma\left(\sum_{k=0}^K T_k(\tilde{\mathbf{L}}^{(t)})X_t\mathbf{\Theta}_k\right)$ (Eq. (11))
6: **end for**
7: Construct dataset $\mathcal{D} = \{\mathbf{h}_i\}$ from all $\mathbf{H}^{(t)}$
8: Train SimCLR encoder $f_{\text{MLP}}$ with contrastive loss $\mathcal{L}_{\text{contrastive}}$ (Eqs. (14), (15))
9: Obtain node latent embeddings $\mathbf{z}_i = f_{\text{MLP}}(\mathbf{h}_i)$ (Eq. (13))
10: **for** each input sample $\mathbf{x}$ **do**
11:     Compute expert outputs $\mathbf{e}_m = E_m(\mathbf{x})$, $m = 1, \ldots, M$ (Eq. (16))
12:     Compute gating weights $\boldsymbol{\alpha} = G(\mathbf{x})$ (Eq. (17))
13:     Aggregate representation $\mathbf{z} = \sum_{m=1}^M \alpha_m \mathbf{e}_m$
14:     Predict anomaly probability $\hat{\mathbf{y}} = \text{softmax}(C(\mathbf{z}))$ (Eq. (18))
15: **end for**
16: Train MoE detector by minimizing cross-entropy loss $\mathcal{L}$ (Eq. (19))
17: **return** $\{\hat{y}_v^{(t)}\}$ anomaly predictions for all nodes in new timestamp

---

Table 1 shows the overall algorithm workflow of CHEBMOE. The process contains three stages:

- **Chebyshev Feature Extraction:** Compute spectral node embeddings from the normalized graph Laplacian (Algorithm steps 1–6).

- **Contrastive Anomaly Generation:** Encode anomalous node features with contrastive learning to improve feature discrimination (Algorithm step 7).

- **Mixture of Experts Detection:** Dynamically aggregate multiple expert networks for final anomaly prediction (Algorithm steps 8–13).

The CHEBMOE framework processes dynamic graph snapshots $\mathcal{G} = \{G_t = (V_t, E_t, X_t)\}_{t=1}^T$ in three main stages: Chebyshev feature extraction, Contrastive anomaly generation, and Mixture-of-Experts (MoE) detection. In the first stage, for each graph snapshot $G_t$, the normalized graph Laplacian $\mathbf{L}^{(t)}$ is computed (Eq. (1)) and scaled to $\tilde{\mathbf{L}}^{(t)}$ (Eq. (4)) to bound the spectrum. Chebyshev polynomial filters $T_k(\tilde{\mathbf{L}}^{(t)})$ for $k = 0, \ldots, K$ are then applied to extract spectral node features $\mathbf{H}^{(t)} = \sigma\left(\sum_{k=0}^K T_k(\tilde{\mathbf{L}}^{(t)})X_t\mathbf{\Theta}_k\right)$ (Eq. (11)), encoding multi-hop structural information without explicit eigendecomposition. Next, all spectral features from the snapshots are collected into a dataset $\mathcal{D} = \{\mathbf{h}_i\}$. A contrastive encoder $f_{\text{MLP}}$ is trained on $\mathcal{D}$ using contrastive loss $\mathcal{L}_{\text{contrastive}}$ (Eqs. (14), (15)) to obtain discriminative node embeddings $\mathbf{z}_i = f_{\text{MLP}}(\mathbf{h}_i)$ (Eq. (13)) suitable for anomaly detection. Finally, the Mixture-of-Experts (MoE) detector processes each node embedding $\mathbf{x}$ by computing expert outputs $\mathbf{e}_m = E_m(\mathbf{x})$ and gating weights $\boldsymbol{\alpha} = G(\mathbf{x})$. The aggregated representation $\mathbf{z} = \sum_{m=1}^M \alpha_m \mathbf{e}_m$ is then fed into a classifier $C$ to predict the anomaly probability $\hat{\mathbf{y}} = \text{softmax}(C(\mathbf{z}))$ (Eq. (18)). The MoE detector is trained end-to-end by minimizing cross-entropy loss $\mathcal{L}$ (Eq. (19)), producing anomaly predictions $\{\hat{y}_v^{(t)}\}$ for all nodes in new timestamps.

# E TIME COMPLEXITY ANALYSIS OF CHEBMOE

In this section, we provide a clear analysis of the computational complexity for each major component of CHEBMOE.

**Chebyshev Feature Extractor** At each time step, the Chebyshev feature extractor applies polynomial filters of order $K$ to a graph with $N$ nodes and $E$ edges. The dominant cost comes from

repeated sparse matrix multiplications, resulting in a per-step complexity of:

$$O\big(K \cdot E \cdot d \cdot F\big) \tag{52}$$

where $d$ is the input feature dimension and $F$ is the number of output filters. The use of sparse operations ensures scalability with respect to the number of edges $E$.

**Contrastive Anomaly Feature Generator** This module processes features of anomalous nodes (each of dimension $F$) in batches of size $B$. The forward pass through the MLP encoder has a complexity of:

$$O(B \cdot F \cdot p) \tag{53}$$

where $p$ is the hidden layer size. The computation of the contrastive loss, which involves all pairwise similarities within the batch, requires:

$$O(B^2 \cdot p) \tag{54}$$

**Mixture of Experts Anomaly Detector** The Mixture of Experts (MoE) detector consists of $M$ expert subnetworks, each incurring a cost of $O(C)$ per input sample, where $C$ is the cost of a single expert. The total cost per sample is:

$$O(M \cdot C) \tag{55}$$

The additional cost of the gating network is negligible compared to the experts.

**Overall Complexity** Aggregating all components over $T$ time steps, the total computational complexity of CHEBMOE is:

$$O\big(T \cdot (K \cdot E \cdot d \cdot F + M \cdot C)\big) \tag{56}$$

This expression highlights that the model scales linearly with the number of time steps and is efficient for large, sparse graphs due to the reliance on sparse matrix operations.

# F    ADDITIONAL EXPERIMENTAL DETAILS

*(a) Datasets.* We evaluate our model on a diverse collection of real-world dynamic graph datasets, each exhibiting unique structural and temporal characteristics that pose significant challenges for anomaly detection:

**(1) Wikipedia (Kumar et al., 2019).** Captures the evolving hyperlink network of Wikipedia articles, reflecting temporal editing and linking behaviors.

**(2) Reddit (Kumar et al., 2019).** Contains temporal user interaction graphs from Reddit subreddits, modeling dynamic social engagement patterns.

**(3) EU-Core1** and **EU-Core3 (Guo et al., 2022).** Represent dynamic collaboration networks among EU researchers and institutions.

**(4) ALPHA (Kumar et al., 2016).** Cryptocurrency transaction networks from the Bitcoin ecosystem, containing highly dynamic financial behavior.

**(5) UCI (Zheng et al., 2019).** Aggregates several dynamic graphs from the UCI Machine Learning Repository, forming a heterogeneous benchmark.

**Note.** All datasets are highly imbalanced, with less than 5% of nodes labeled as anomalous, aligning with real-world anomaly detection scenarios such as fraud detection and intrusion monitoring. This imbalance is critical for evaluating model robustness in practical applications.

*(b) Baselines.* We compare our method with seven state-of-the-art approaches for dynamic graph anomaly detection:

**(1) TGAT (Xu et al., 2020).** A temporal graph attention network that learns time-aware node representations.

**(2) DOMINANT (Ding et al., 2019).** Combines structural and attribute reconstruction losses for unsupervised anomaly detection.

**(3) DONE (Bandyopadhyay et al., 2020).** Integrates GCNs and temporal autoencoders for capturing evolving graph behaviors.

**(4) CONAD (Xu et al., 2022).** Employs contrastive learning with temporal augmentation to improve robustness.

**(5) AnomalyDAE (Fan et al., 2020).** Uses deep autoencoders to detect anomalies in dynamic graphs.

**(6) SAD (Tian et al., 2023).** A semi-supervised framework leveraging limited labels to guide anomaly detection.

**(7) MAMF (Hong et al., 2025).** A multitask graph anomaly detection model that leverages meta-learning to improve generalization under scarce anomaly samples. It fuses multiple structural features and learns task-agnostic representations, enabling the model to adapt quickly to new temporal slices or unseen anomaly patterns.

*(c) Evaluation Metrics.* To comprehensively evaluate anomaly detection performance, we adopt four standard metrics:

**(1) ROC-AUC (Receiver Operating Characteristic - Area Under Curve):** Quantifies the model's ability to distinguish between anomalous and normal nodes across all possible thresholds. It is based on the True Positive Rate (TPR) and False Positive Rate (FPR):

$$\text{TPR} = \frac{\text{TP}}{\text{TP} + \text{FN}}, \quad \text{FPR} = \frac{\text{FP}}{\text{FP} + \text{TN}}, \tag{57}$$

where $\text{TP}$, $\text{FP}$, $\text{TN}$, and $\text{FN}$ denote true positives, false positives, true negatives, and false negatives, respectively.

**(2) Precision:** Measures the proportion of correctly identified anomalous nodes among all nodes predicted as anomalies:

$$\text{Precision} = \frac{\text{TP}}{\text{TP} + \text{FP}}. \tag{58}$$

**(3) F1-score:** The harmonic mean of Precision and Recall, providing a balanced measure of accuracy and completeness:

$$\text{F1} = 2 \times \frac{\text{Precision} \times \text{Recall}}{\text{Precision} + \text{Recall}}, \tag{59}$$

where Recall is defined as $\frac{\text{TP}}{\text{TP}+\text{FN}}$.

**(4) AUPR (Area Under Precision-Recall Curve):** Summarizes the trade-off between Precision and Recall across different thresholds, and is especially informative for highly imbalanced datasets:

$$\text{AUPR} = \int \text{Precision}(\text{Recall}) \, d(\text{Recall}). \tag{60}$$

## G  ADDITIONAL EXPERIMENTAL RESULTS

In this section, we provide additional experimental results to complement the main results.

### G.1  FREQUENCY RESPONSE CURVES ON ADDITIONAL DATASETS

To further support our main findings, we present frequency response curves for additional datasets in Figure 6. These results clearly show that Chebyshev filters consistently retain high-frequency spectral components—features that are often indicative of anomalies—across all datasets. In contrast, GCN filters tend to suppress these high-frequency signals, highlighting the spectral washing effect and underscoring the advantage of Chebyshev-based filtering for anomaly detection.

**Observation.** For all datasets, Chebyshev-based filters consistently preserve significant high-frequency components, which are crucial for effective anomaly detection. In contrast, GCN filters strongly suppress high-frequency signals, illustrating the spectral washing phenomenon as shown in Figure 6.

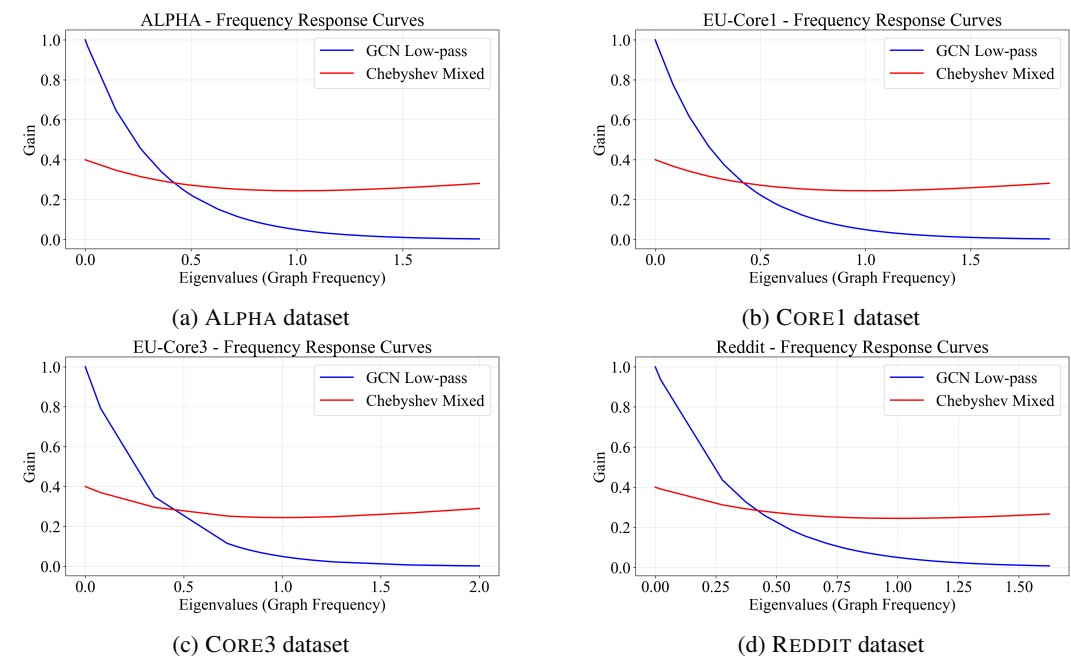

(a) ALPHA dataset        (b) CORE1 dataset

(c) CORE3 dataset        (d) REDDIT dataset

Figure 6: Frequency response curves for the remaining datasets. Chebyshev filters preserve anomaly-relevant high-frequency components across all datasets, whereas GCN filters suppress them.

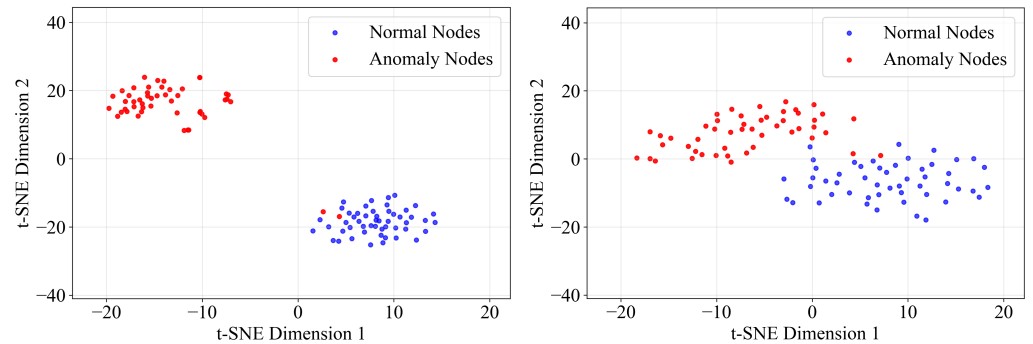

Figure 7: t-SNE visualization of normal (blue) and anomalous (red) nodes on ALPHA dataset. Left: with mitigation; Right: without mitigation. Spectral washing mitigation improves anomaly separability.

## G.2    T-SNE VISUALIZATION ON OTHER DATASETS

In Section 5a, we presented a representative t-SNE visualization on the UCI dataset. For completeness, we include additional visualizations on the remaining datasets: ALPHA (Figure 7), EU-CORE1 (Figure 8), EU-CORE3 (Figure 9), WIKIPEDIA (Figure 10), and REDDIT (Figure 11). Across all datasets, mitigating spectral washing consistently improves the separation between anomalous and normal nodes. Without mitigation, anomalies remain partially distinguishable but are less clearly separated, confirming the robustness of our spectral mitigation approach.

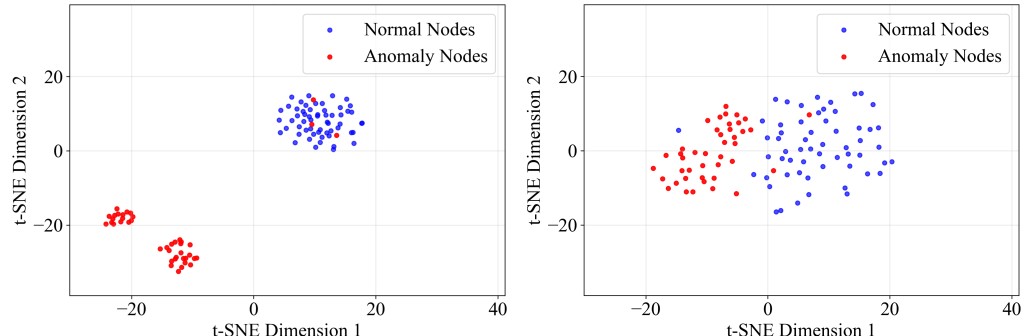

Figure 8: t-SNE visualization on EU-CORE1 dataset. Left: with mitigation; Right: without mitigation. The separation between anomalies and normal nodes is enhanced with spectral mitigation.

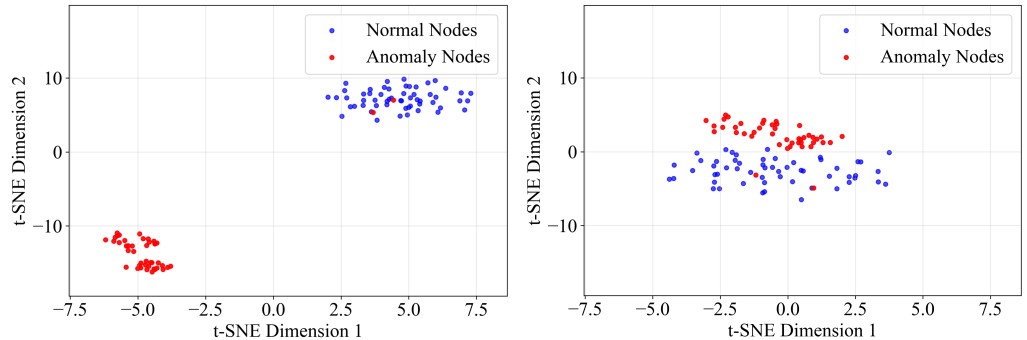

Figure 9: t-SNE visualization on EU-CORE3 dataset. Left: with mitigation; Right: without mitigation. Spectral washing mitigation makes anomalies more distinct.

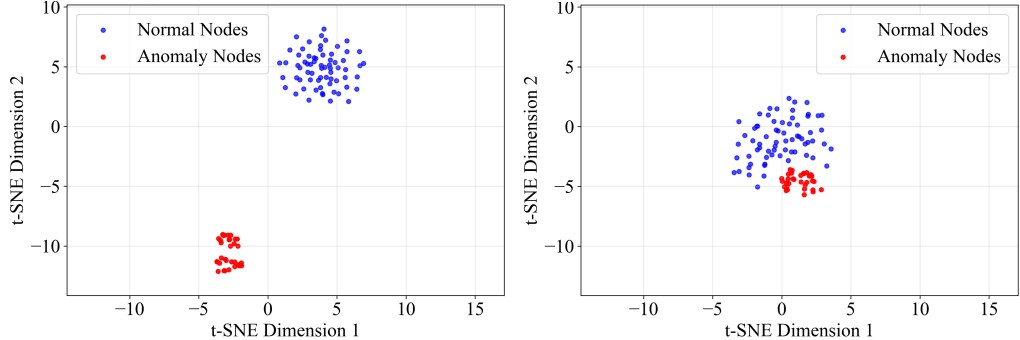

Figure 10: t-SNE visualization on WIKIPEDIA dataset. Left: with mitigation; Right: without mitigation. The Chebyshev-based approach clearly separates anomalous nodes from normal nodes.

## H    ADDITIONAL PARAMETER ANALYSIS

We provide a detailed study of three hyper-parameters in CHEBMOE: ChebNet order $K$, number of experts $M$, and SimCLR temperature $\tau$. Results are reported across four representative datasets using four metrics: AUPR, F1-score, Precision, and ROC-AUC.

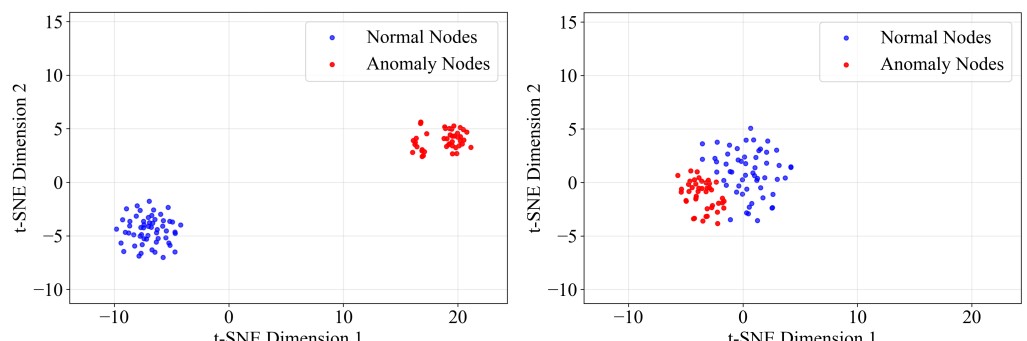

Figure 11: t-SNE visualization on REDDIT dataset. Left: with mitigation; Right: without mitigation. Spectral washing mitigation consistently improves anomaly separability across datasets.

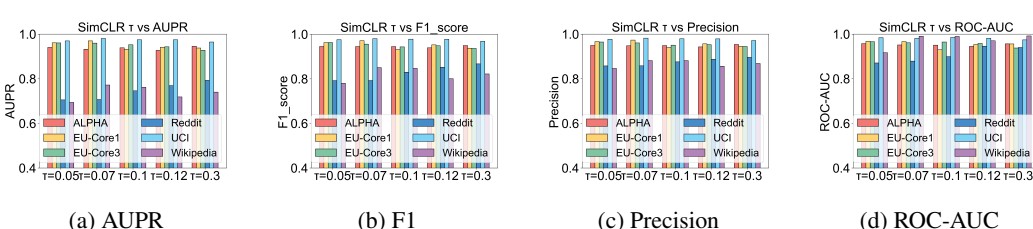

(a) AUPR        (b) F1        (c) Precision        (d) ROC-AUC

Figure 12: Sensitivity of SimCLR temperature $\tau$.

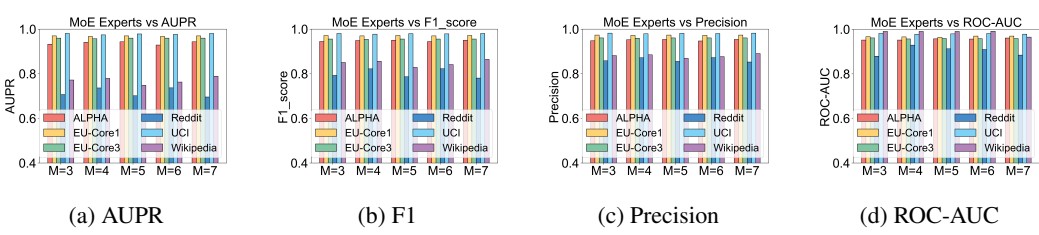

(a) AUPR        (b) F1        (c) Precision        (d) ROC-AUC

Figure 13: Impact of the number of experts $M$.

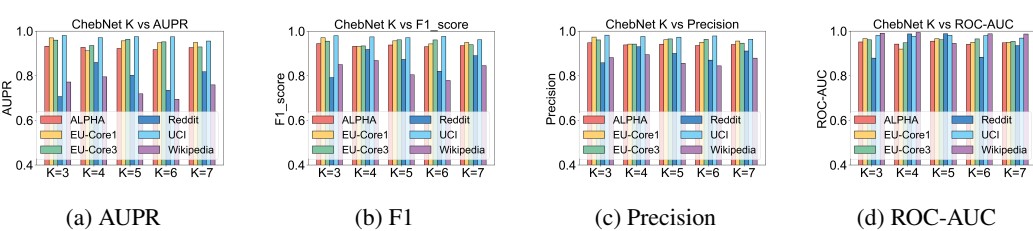

(a) AUPR        (b) F1        (c) Precision        (d) ROC-AUC

Figure 14: Sensitivity of ChebNet order $K$.

