# OpenReview forum: "ChebMoE: A Spectral-Aware and Expert-Adaptive Framework for Graph Anomaly Detection"
_ICLR.cc/2026/Conference — ICLR 2026 Conference Withdrawn Submission_

### Official Review · Reviewer_fhdD · 2025-10-18

**Soundness:** 2
**Presentation:** 3
**Contribution:** 2
**Rating:** 2
**Confidence:** 4

**Summary:**

The authors present CHEBMOE, a framework that tries to address the spectral limitations of existing GNN-based anomaly detectors for dynamic graph anomaly detection. Experiments show the effectiveness to some extent.

**Strengths:**

1. The authors present CHEBMOE, a framework that tries to address the spectral limitations of existing GNN-based anomaly detectors for dynamic graph anomaly detection.
2. Experiments show the effectiveness to some extent.

**Weaknesses:**

1. The novelty is limited. The right shift of spectral space is already utilized in several previous works in the graph anomaly detection field, which means the observation can not be a contribution. Besides, the design of the framework is a combination of exiting work, instead of a carefully designed one for the observation, further degrading the novelty of the work.
2. The experimental comparison is not convincing. Since the training objective of the MoE part requires ground truth labels, the comparison with several unsupervised frameworks can be unfair. Furthermore, most of the compared baselines are relatively old. The authors should consider new baselines in this area, such as [1-4].
3. The hyperparameters can lead to instability of the framework. As shown in Figures 12, 13, and 14, the performance of some datasets can vary significantly when changing the hyperparameters, which raises the question of how to choose appropriate hyperparameters for new examples without tedious grid search.

[1] Xiao Yang, Xuejiao Zhao, Zhiqi Shen. A Generalizable Anomaly Detection Method in Dynamic Graphs. AAAI 2025.

[2] Yifan Hong, Muhammad Asif Ali, Huan Wang, Junyang Chen, Di Wang. ABNet: Mitigating Sample Imbalance in Anomaly Detection Within Dynamic Graphs. IJCAI 2025.

[3] Jianhao Guo, Siliang Tang, Juncheng Li, Kaihang Pan, Lingfei Wu. RustGraph: Robust Anomaly Detection in Dynamic Graphs by Jointly Learning Structural-Temporal Dependency. TKDE 2024.

[4] Dong Chen, Xiang Zhao, Weidong Xiao. Fine-grained Anomaly Detection on Dynamic Graphs via Attention Alignment. ICDE 2024.

**Questions:**

Please refer to the weaknesses.

---

### Official Review · Reviewer_1SE7 · 2025-10-27

**Soundness:** 2
**Presentation:** 2
**Contribution:** 2
**Rating:** 4
**Confidence:** 4

**Summary:**

This paper proposes Chebyshev-based Network with Mixture-of-experts, namely ChebMoE, a spectral-aware and expert-adaptive method for graph anomaly detection that addresses the “spectral washing” problem in conventional GNNs, where low-pass filtering suppresses high-frequency anomaly signals. ChebMoE integrates Chebyshev polynomial filtering to preserve spectral information, contrastive learning to enhance anomaly feature representation, and a mixture-of-experts mechanism to adaptively detect diverse anomalies. Theoretical analysis supports its spectral design, and experiments are conducted on several real-world dynamic datasets to evaluate the proposed method.

**Strengths:**

1. This paper is well-motivated, which clearly identifies a meaningful limitation of existing GNN-based anomaly detectors that the loss of high-frequency anomaly information due to spectral smoothing.

2. The proposed CHEBMOE is theoretically sound, it uses Chebyshev polynomial filters to approximate spectral responses without requiring eigendecomposition. Meanwhile, the integration of spectral modeling, contrastive learning, and the mixture-of-experts design is creative, well-motivated, and technically consistent.

3. This paper presents experiments across multiple benchmark datasets, demonstrating extensive experimental coverage.

**Weaknesses:**

1. The paper is not well written, it is hard to follow at times. For example, in lines 278-279, it is stated that “We select features of nodes flagged as anomalous, which serve as input to the contrastive generator.” , but it is not explicit what these “flags” are.

2. Results are based on 20-run averages but tables lack standard deviations or confidence intervals , making it difficult to assess stability or whether reported improvements exceed random fluctuations.

3. Missing direct comparisons to other spectral-aware filters in ablation study.

**Questions:**

1. Is the contrastive encoder trained separately (pretrain) or jointly end-to-end with the MoE classifier?

2. How to generate two augmentations per node? In Fig 2, two augmentations are enhanced first and then used as the input for MLP. However, when described in the text, the augmentations seem to be derived from the embedding after MLP.

3. Is the SimCLR encoder trained only on the subset labeled as anomalies, or on all nodes to construct a good discriminative space?

---

### Official Review · Reviewer_Bo1a · 2025-10-28

**Soundness:** 2
**Presentation:** 2
**Contribution:** 2
**Rating:** 2
**Confidence:** 4

**Summary:**

This paper proposes ChebMoE, a framework for graph anomaly detection that combines Chebyshev polynomial-based spectral filtering, contrastive learning, and mixture-of-experts (MoE) architecture. While the paper addresses the important problem of spectral washing in GNN-based anomaly detection, it suffers from significant weaknesses in novelty, experimental evaluation, and theoretical rigor.

**Strengths:**

1. Clear problem identification: The paper clearly articulates the spectral washing phenomenon where standard GNNs suppress high-frequency components critical for anomaly detection.

2. Comprehensive theoretical appendix: Appendix B provides detailed mathematical formulations of spectral washing effects.

3. Consistent empirical improvements: The method shows performance gains across multiple datasets.

**Weaknesses:**

1. The paper's main contribution amounts to a relatively rigid combination of existing techniques without deep integration:
- Chebyshev filtering has been extensively studied for graph signal processing (Zhou 2010; Shuman et al. 2018; Defferrard et al. 2016's ChebNet). The application here is straightforward without novel theoretical insights specific to anomaly detection.
- Contrastive learning via SimCLR/InfoNCE is standard practice (Chen et al. 2020). The "contrastive anomaly feature generator" (Section 3.2) simply applies existing contrastive loss to anomaly features without addressing why this specific combination is necessary.

The paper essentially stacks these modules (Chebyshev → Contrastive → MoE → Classifier) without demonstrating why this particular pipeline is superior to alternatives or how these components synergistically address the core spectral washing problem. The modules solve different problems (spectral preservation, feature augmentation, adaptive detection) rather than being designed as an integrated solution to spectral washing.


2. The paper has a conspicuous omission of BWGNN (Tang et al., 2022), which directly addresses high-frequency signal preservation for anomaly detection using adaptive spectral filtering. This is particularly problematic because: BWGNN explicitly tackles the same spectral shift problem cited by this paper (Tang et al., 2022 is referenced for defining spectral shift); The absence of BWGNN from baselines undermines the credibility of performance comparisons

3. Missing GADBench methods: The paper omits recent strong baselines from GADBench (Tang et al., 2023), a comprehensive benchmark suite specifically for graph anomaly detection. This selective baseline comparison raises concerns about cherry-picking to favor the proposed method.

4. No fine-grained ablations: What happens with ChebNet alone without MoE? What is the individual contribution of contrastive learning to the final performance?

5. No formal guarantee that Chebyshev filtering improves anomaly detection: Corollary 7 provides an energy preservation bound, but does not establish that preserved high-frequency energy translates to better anomaly detection performance

6. The method processes each snapshot G_t independently (Algorithm 1, lines 1-6). How is temporal information leveraged? The paper claims to address "dynamic graphs," but the temporal modeling is unclear

**Questions:**

1. Why is BWGNN not included as a baseline, given its direct relevance to spectral preservation for anomaly detection?

2. What is the computational overhead of your method compared to baselines? Please provide wall-clock time comparisons.

3. How does the method perform when anomalies do not exhibit high-frequency characteristics (e.g., low-degree anomalous nodes in dense regions)?

4.

---

### Official Review · Reviewer_3rxb · 2025-10-29

**Soundness:** 2
**Presentation:** 2
**Contribution:** 2
**Rating:** 2
**Confidence:** 4

**Summary:**

This paper addressed spectral washing in graph anomaly detection, where prior works suppressed high-frequency signals critical for identifying anomalies. The authors proposed CHEBMOE, a three-component framework: (1) Chebyshev-based Feature Extractor (ChebNet) to preserve high-frequency spectral information; (2) Contrastive Anomaly Feature Generator (CAFG) to address the scarcity of anomaly labeled samples and (3) Mixture-of-Experts Anomaly Detector (MoE-AD) to adaptively capture evolving temporal anomaly patterns.

**Strengths:**

(S1) Insightful problem observation and targeted solution: The paper identified a limitation in GNN-based anomaly detection. The low-pass nature of GCN suppressed high-frequency anomaly signals. The adoption of ChebNet directly targeted this issue by preserving high-frequency components.

**Weaknesses:**

(W1) The core problem to be resolved and the proposed method are inconsistent. The abstract and introduction focus on spectral washing as the primary challenge, with brief mention of temporally dynamic anomalies. However, when introducing the proposed framework, a third issue--scarcity of labeled anomalies--suddenly appeared to justify the design of CAFG, despite never being discussed earlier. This creates a narrative disconnect: the paper established spectral washing as the central problem but proposed a three-component solution where two components (CAFG and MoE-AD) address issues (label scarcity and temporal dynamics) that are inadequately motivated upfront.

(W2) The methodology design lacks novelty and customization to graph data. The proposed components are largely direct adoptions of existing techniques without innovation or graph-specific adaptations. The Chebyshev Feature Extractor is essentially ChebNet without modifications; CAFG basically follows the SimCLR pipeline; and MoE-AD is a simple adoption of vanilla MoE architecture. Critically, both CAFG and MoE-AD fail to leverage graph-specific characteristics: CAFG treated node embeddings as independent vectors, ignoring structural relationships, while MoE-AD operated without any customization for graph topology.

(W3) The writing lacks clarity. For example, the description of MoE-AD’s input is ambiguous. Based on Figure 2 and the provided source code, MoE-AD appeared to take two types of inputs: (1) embeddings from the ChebNet and (2) embeddings from the CAFG. However, Section 3.3 fails to explicitly clarify this dual-input design. The authors vaguely introduced the input as "x" without specifying where "x" comes from.

(W4) The provided code is incomplete for reproducibility. It only provided methodology scripts while missing several critical components: (1) dataset preprocessing pipeline scripts, (2) environment specifications (e.g., requirements.txt), and (3) README file with proper documentation. These omissions significantly hinder reproducibility.

**Questions:**

1. Is the Chebyshev-based Feature Extractor a direct adoption of ChebNet, or are there any differences between them?

2. What is the input to MoE-AD? What is the architect of experts in MoE?

3. Are there any customized designs in CAFG and MoE-AD that are specifically tailored for graph data?

4. Since the paper framed "high-frequency removal" as an underexplored core issue in graph anomaly detection, why not addressing it in static graph settings where MoE (a simple adoption) would be unnecessary? If the authors mainly aimed to resolve "high-frequency removal", a static setting seems adequate; introducing dynamic graphs only adds unnecessary complexity without deepening the core contribution.

---

### Comment · Area_Chair_1GYk · 2025-11-28

Dear Reviewers,

Thank you for your valuable time and expertise in reviewing this paper.

The authors have now submitted their rebuttal. We would appreciate it if you could review their responses and assess whether your concerns have been addressed, if you haven't done this.

Best regards,

AC

---

> ### Comment · Reviewer_3rxb · 2025-11-28
> **Inquiry about rebuttals**
>
> Dear AC,
>
> I have not found any rebuttals yet. Could you cordially make sure whether the authors have submitted their rebuttals to the system?

---

> > ### Comment · Area_Chair_1GYk · 2025-11-28
> >
> > Dear Reviewers,
> >
> > Please accept our apologies for this error. The authors have not submitted a rebuttal yet.
> >
> > Thank you for your attention.
> >
> > Best regards,
> >
> > AC

---

### Note · Authors · 2025-11-28

I have read and agree with the venue's withdrawal policy on behalf of myself and my co-authors.